# BLI-Based Functional Assay in Phage Display Benefits the Development of a PD-L1-Targeting Therapeutic Antibody

**DOI:** 10.3390/v12060684

**Published:** 2020-06-25

**Authors:** Jong Rip Choi, Min Jung Kim, Nara Tae, Tae Min Wi, Se-Ho Kim, Eung Suk Lee, Dae Hee Kim

**Affiliations:** 1Scripps Korea Antibody Institute, KNU Chuncheon Campus, Chuncheon, Gangwon 200-701, Korea; cjr@skai.or.kr (J.R.C.); kmj@skai.or.kr (M.J.K.); nalgoon@gmail.com (N.T.); wtm9272@naver.com (T.M.W.); 2Department of Systems Immunology, College of Biomedical Science, Kangwon National University, Chuncheon, Gangwon 200-701, Korea; kimsho@kangwon.ac.kr

**Keywords:** phage display, biolayer interferometry (BLI), scFv, high-throughput screening, antibody, PD-L1

## Abstract

The therapeutic functionality of the antibodies from phage display is verified after an initial screening. Several immunological assays such as ELISA, flow cytometry, the western blot, and surface plasmon resonance (SPR) assay are commonly used; the IgG-format antibody is usually preferred to verify the functionality of antibodies, which need elaborative mammalian expression and purification work. Here, we describe a biolayer interferometry (BLI)-based assay that can evaluate the inhibitory functions of antibodies at an earlier stage of screening. To develop a PD-L1-targeting antibody from phage display, we applied the BLI assay to the initial scFv antibody screening, in addition to common ELISA and fluorescence-activated cell sorting (FACS) assays, which showed high advantages and relevance with the in vitro cell-based PD-1/PD-L1 inhibition assay. The same assays for IgG-format antibodies showed high efficiency of the BLI assay in the functional characterization of antibodies, and one candidate selected from the BLI assay resulted in highly efficacious antitumor activity in an in vivo syngeneic mouse study. The BLI assay was also beneficial when searching for antibodies with diverse epitopes. These results demonstrated that the BLI-based inhibition assay is an excellent technique for high-throughput scFv antibody screening in earlier stages and can make phage-display antibody screening more efficient to develop therapeutic candidates.

## 1. Introduction

Programmed death-ligand 1 (PD-L1) is highly expressed in diverse cancers, and is regarded as a major contributor to the immune incompetency of immune cells in cancer microenvironments through the ligation between PD-L1 and programmed cell death protein 1 (PD-1) on T cells that generate strong suppressive signals in the activated T cell to make it anergic [1,2,3,4,5]. After the reports in the *New England Journal of Medicine* in June 2012 on the phase I studies of anti-PD-L1 and PD-1 antibodies amazed the whole oncology field [6,7], immune checkpoint inhibitory antibodies have become some of the most promising antibody drugs to treat several cancers in the clinic [8]. The reinvigoration of immune reactivity by PD-1 and PD-L1 targeting antibodies toward several cancers has been quite successful; therefore, the enhanced usage of these antibodies as a “backbone” regimen of anticancer combination treatments is likely [9].

“Phage display” was initially described by Smith in 1985 [10], in which the coating proteins of filamentous bacteriophages fused with exogenous peptides were expressed on their surface and used for the enrichment of specific phages. Subsequently, Adalimumab (Humira) was developed in 1990 by Winter as the first phage-display-derived therapeutic antibody to neutralize tumor necrosis factor-alpha (TNF-α) for immunological disease treatment. Antibody phage-display technologies were further developed by several other groups (Winter and McCafferty, Cambridge, UK; Lerner and Barbas, California, USA; and Breitling and Dübel, Heidelberg, Germany) [11,12,13,14,15,16]. In 2002, Adalimumab (Humira) was approved by the FDA for the treatment of rheumatoid arthritis and is currently the best-selling drug in the world [17]. In 2018, Smith and Winter were jointly awarded the Nobel Prize in chemistry for their contributions to the development of the phage-display technology.

Phage display became the most powerful and commonly used technology for antibody drug development and has been used in various other research areas, including enzyme optimization, affinity maturation of antibodies, the development of nanovehicles (nanomedicines), epitope mapping, vaccine development, and biomolecular interaction studies [18,19,20,21,22,23,24]. For antibody screening using phage display, a single-pot antibody library is applied to target antigens and 3‒4 rounds of panning are usually performed to enrich the antibody binders. At the end of panning, bacterial colonies infected with individual phages are used for the characterization of each clone’s binding to an antigen. Usually, phage particles or scFv/Fab are used for the first binding assessment of each antibody without detailed normalization information [25]; the antibody conversion to IgG format commonly follows. These IgG-format purified antibodies are applied to most antibody characterizations, such as enzyme-linked immunosorbent assay (ELISA), fluorescence-activated cell sorting (FACS) analysis on cell surface antigens, direct protein interaction analysis through surface plasmon resonance (SPR), and other in vitro cell-based assays, through which functional antibody candidates can usually be selected.

However, this typical phage-display process requires a lot of effort and resources to get the functional antibody candidates in the end; IgG conversion and antibody production and purification from a mammalian cell system in large quantities, especially, are difficult steps for academic researchers [26,27]. Therefore, the elimination of unqualified antibody candidates at an earlier stage and earlier functional characterization of antibody clones will speed up the selection of highly functional candidates and result in more successful therapeutic antibody development.

Biolayer interferometry (BLI) is used to characterize direct antibody‒antigen interactions, especially in the Octet system from Pall ForteBio (Fremont, CA, USA) [28]. BLI is a label-free technique that enables real-time monitoring of biomolecule interactions by analyzing optical interference pattern changes on protein-coated sensor chip surfaces generated by mass increases from analyte‒protein interactions. Its mechanism, basically the same as the surface plasmon resonance effect, is taken into account by the well-known Biacore analytical system from Cytiva (Marlborough, MA, USA). However, 96-well or 384-well BLI assay facilitates high-throughput analysis, which offers greater potential in drug development applications.

In this study, we designed a new BLI-based functional scFv screening method and applied it to screen out functional antibodies targeting the PD-L1 immune-checkpoint ligand protein. The BLI assay in an earlier stage showed its high functional relevance with an in vitro cell-based assay and that extended well to an in vivo syngeneic mouse model study. Our study showed that the BLI-based assay is highly beneficial for functional antibody screening and drug candidate development.

## 2. Materials and Methods

### 2.1. Materials and Reagents

Human and mouse PD-L1 cDNA were purchased from Sino Biologicals, Inc. (Beijing, China). PCR was performed using the AccuPrime pfx DNA polymerase (Invitrogen, Carlsbad, CA, USA). All restriction enzymes and T4 ligase were purchased from New England Biolabs (Ipswich, MA, USA). Recombinant mouse interferon gamma (IFN-γ) was purchased from Pepro Tech (Rocky Hill, NJ, USA). 1-Ethyl-3-(3-dimethylaminopropyl)carbodiimide hydrochloride (EDC), N-hydroxysuccinimide (NHS), ethanolamine, and Amine Reactive Second-Generation (AR2G) biosensors were obtained from ForteBio (Fremont, CA, USA).

### 2.2. Cell Lines and Transfection

PC3 and 293T cells were obtained from the American Type Culture Collection (Manassas, VA, USA). The murine colon cancer cell line MC38 was a kind gift from Dr. Sin, Kangwon National University (Chuncheon, Korea). Cells were cultured in RPMI-1640 or DMEM (Gibco, Grand Island, NY, USA), supplemented with 10% *v*/*v* fetal bovine serum (FBS) and 1% *v*/*v* penicillin/streptomycin (Gibco). FreeStyle 293F cells were maintained in Freestyle 293 expression medium (Invitrogen). The 293T and FreeStyle 293F cells were transfected using Lipofectamine 2000 (Invitrogen) or FectoPRO transfection reagent (Polyplus-transfection, Illkirch, France).

### 2.3. Phage Panning

For our phage-display antibody screening, a human naïve antibody library kindly provided by LG Life Sciences, a private biotech company in South Korea, through a material transfer agreement was used to isolate human antibody clones binding to human PD-L1. The antibody library has about 1.2 × 10^11^ diversity and each scFv is expressed with a c-Myc tag at its C-terminus. The human naïve scFv library was re-amplified as scFv-displaying phages and the phage pools from each round of panning were generated using ER2738 cells (New England Biolabs). Phage-display experiments including library reamplification and phage panning were mainly performed with the protocols described in *Phage Display: A Laboratory Manual* [29], with the following additional details.

To rescue scFv-displaying phages, frozen library stocks were thawed and inoculated into super broth (SB) medium (containing 2% glucose, 35 μg/mL chloramphenicol). A VCSM13 helper phage (10^12^ colony-forming units) was then added to the library cells in the exponential growth phase. After overnight culture, the phage was purified by PEG precipitation (final 5% *w*/*v* polyethylene glycol 8000 (PEG8000), 0.5 M NaCl).

Biopanning for the enrichment of scFv binders was performed using human IgG (12.5 μg/mL, Green Cross Corp., Yong-in, Korea)-coated immunotubes (Nunc, Roskilde, Denmark) or PD-L1-conjugated magnetic beads (4 μg/10^7^ beads, Dynabeads M270 epoxy, Invitrogen, Carlsbad, CA, USA). In order to remove nonspecific scFv phage binders from the Fc region, a subtraction reaction was performed using a human IgG-coated immunotube for 2 h at room temperature, then the scFv phages in the supernatant were incubated for 1 h at 4 °C with PD-L1-conjugated magnetic beads. The phage-PD-L1 antigens were washed one time for the first panning, 2‒3 times for the second and third panning, and 3‒5 times for the fourth panning with PBS (containing 0.1% Tween 20) to introduce increased stringency and obtain high-affinity binders. Bound phages were eluted with 0.25% trypsin (Gibco). Eluted phages were amplified by infection to *E. coli* ER2738 and the phage rescued as outline above. A total of four rounds of panning were performed to enrich the scFv antibody binders and individual colonies picked randomly from the third and fourth rounds of panning were sought after the target binding assays.

### 2.4. scFv Expression and TES Periplasmic Extraction

Single colonies expressing individual scFvs from the third and fourth panning pools were randomly picked and grown overnight in a 96-well plate containing 300 μL/well of super broth (SB) with carbenicillin. Twenty microliters of overnight-grown cultures were inoculated in 900 μL/well of SB medium carbenicillin and cultured at 30 °C with shaking. When the absorbance at 600 nm reached 0.7, IPTG was added to each well (1 mM final concentration) and cells were grown overnight at 30 °C with shaking. After tabletop centrifugation, a periplasmic extract was obtained using the osmotic shock method. Briefly, after centrifugation at 3500 rpm at room temperature (to discard the culture media), the cell pellets were resuspended in 320 μL of ice-cold extraction solution (TES buffer, containing 20% *w*/*v* sucrose, 50 mM Tris, 1 mM EDTA, pH 8.0) and after 5 min on ice bath, 480 μL of ice-cold 0.2 × TES buffer was added to each well. After 30 min incubation on ice, the suspension was centrifuged at 3500 rpm, 4 °C, and scFv-containing periplasmic crude extracts were obtained as the supernatants. scFv-containing periplasmic crude extracts were used for the enzyme-linked immunosorbent assay (ELISA), flow cytometry, and biolayer interferometry (BLI)-based assay.

### 2.5. Expression of IgG Format Antibodies Using a Two-Vector System for Heavy and Light

With germline sequence information for 72 scFv clones, the primers for those VH and VL domains were synthesized and used for PCR amplification of individual VH and VL fragments (Appendix A). Each VH primer was designed to have restriction sites for KpnI or BamHI at the 5′ ends and NheI at the 3′ ends. For the VL primers, BamHI and BsiWI restriction sites were incorporated at the 5′ and 3′ ends, respectively. Amplified VH and VL fragments of each clone were ligated into the pCEP4 mammalian expression vector (Invitrogen) as a fused form, either with CH1-hinge-CH2-CH3 for the heavy chain expression or with C_κ_ for the light chain expression. Matching heavy and light chain vectors for each clone were co-transfected into FreeStyle 293-F cells (Thermo Fisher) with a 2:1 (light vs. heavy) DNA ratio using FectoPRO transfection reagent (Polyplus-transfection) as per the manufacturer’s instructions; after 12 days, the culture medium was harvested. The IgG antibodies were purified by open-column chromatography using protein A agarose beads (GenScript, Piscataway, NJ, USA).

After dialysis with phosphate-buffered saline at pH 7.4 (PBS), the antibody concentration was quantified using a NanoDrop 2000 spectrophotometer (Thermo Fisher Scientific, Waltham, MA, USA), and the purity of each antibody was evaluated by SDS/polyacrylamide gel electrophoresis (PAGE) and Coomassie brilliant blue staining.

### 2.6. ELISA

The specificity and binding activity of individual scFv and IgG were assessed by ELISA. The human PD-L1-Fc and human IgG were applied to a 96-well ELISA plate at a concentration of 10 µg/mL in PBS at 4 °C overnight. After being washed three times with PBS containing 0.05% *v*/*v* Tween 20 (PBST), the wells were blocked with 3% *w*/*v* BSA in PBST for 1 h at 37 °C. Then, 25 μL/well scFv-containing periplasmic crude extracts or 100 ng/well purified IgGs were added and incubated for 1 h at 37 °C. After washing, for scFv ELISA, 2 µg/mL mouse anti-c-Myc IgG (Sigma-Aldrich, St. Louis, MO, USA) was added for 1 h at 37 °C, and, after washing, horseradish peroxidase (HRP)-conjugated goat anti-mouse IgG (1/10,000, Jackson ImmunoResearch, West Grove, PA, USA) in PBST containing 3% BSA was incubated for 1 h at 37 °C. For ELISA using IgG, HRP-conjugated goat anti-human kappa light chain IgG (1:5000, Sigma) was added for 1 h. For detection, 100 μL/well of 3,3’,5,5’-Tetramethyl–benzidine (TMB) substrate (BD Biosciences, San Jose, CA, USA) was used and the absorbance read at 450 nm using a microtiter plate reader (VICTOR X4, Perkin Elmer, Waltham, MA, USA).

### 2.7. Flow Cytometry

The specific binding of the scFv-containing periplasmic crude extracts and purified IgGs directed to cell-surface human PD-L1 and mouse PD-L1 was analyzed by BD FACSCalibur (BD Biosciences). About 5 × 10^5^ cells were used in each experiment. Harvested cells were resuspended in 100 μL FACS buffer (2% BSA and 0.02% NaN3 in PBS) and then mixed with 100 μL scFv-containing periplasmic crude extracts or 10 µg/mL purified IgG in FACS buffer on ice for 1 h. After washing with FACS buffer, cells were treated for 1 h with 2 μg/mL mouse anti-c-Myc antibody and 1 h with Alexa Fluor 488-conjugated goat anti-mouse IgG (1:400, Jackson ImmunoResearch) for periplasmic scFv staining. For flow cytometry using IgG, after 1 h incubation with each IgG clones, Alexa Fluor 488-conjugated goat anti-human IgG (1:400, Jackson ImmunoResearch) was treated for 1 h. Cells were washed between the reactions with a FACS buffer. All samples were measured by a FACSCalibur flow cytometer and CELL QUEST program (BD Biosciences), and the data were analyzed by FlowJo software (FlowJo LLC, Ashland, OR, USA).

### 2.8. BLI-Based PD-1/PD-L1 Binding Inhibition Assay

BLI-based PD-1/PD-L1 binding inhibition experiments were carried out by BLI using an Octet RED96 System (Pall ForteBio). The measurements were performed using amine-reactive second-generation (AR2G) biosensors. The human PD-1-Fc (Sino Biological Inc., Beijing, China) proteins were immobilized on the biosensor tip surface. The binding or dissociation of soluble human PD-L1-Fc proteins at the biosensor tip surface caused a pattern shift in the reflectance interference wave that is correlated with the changes in the mass or thickness of the biolayer at the sensor tip, which is denoted as a sensorgram. All steps were performed at 30 °C with shaking at 400 rpm in a black 96-well plate, with a working volume of 200 μL in each well. The human PD-1-Fc proteins were immobilized onto the AR2G biosensor surface via a 1-Ethyl-3-(3-dimethylaminopropyl)carbodiimide/*N*-hydroxysuccinimide (EDC/Sullfo-NHS)-mediated amine-coupling reaction, in accordance with the manufacturer’s directions. The human PD-1-Fc-immobilized AR2G biosensors were applied to the procedure of baseline (DW, 60 s), sensor activation (EDC/sulfo-NHS, 300 s), 5 µg/mL human PD-1-Fc protein loading (10 mM sodium acetate, pH 5, 1800 s), quenching (1 M ethanolamine, pH 8.5, 300 s), blocking (3% *w*/*v* BSA in PBS, 1800 s), and baseline (running buffer, 120 s). The human PD-L1-Fc protein (10 µg/mL, 100 μL) in the running buffer was mixed with 100 μL scFv-containing periplasmic crude extract, then applied for association for 300 s and dissociation (running buffer, 300 s). For the control sample, the same amount of TES buffer (0.5×, 100 μL) was added to the PD-L1 sample to remove the interference from the TES buffer itself. The response data were normalized using Octet data analysis software version 9.0.0.14 (Pall ForteBio).

### 2.9. PD-1/PD-L1 Blockade Bioassay

The inhibitory activity of the anti-PD-L1 antibody was analyzed by a luciferase reporter assay, performed in accordance with the manufacturer’s instructions (Promega, Fitchburg, WI, USA). PD-L1 aAPC/CHOK1 cells (4 × 10^5^ cells/100 μL/well) in an F12 medium containing 10% FBS were seeded into white 96-well plates (Corning, Corning, NY, USA) and incubated for 17 h at 37 °C. After culturing the PD-L1 aAPC/CHOK1 cells, the medium was removed, and cells were pre-incubated with 10 µg/mL (40 μL/well) anti-PD-L1 antibodies or control antibodies (MK3475 or MPDL3280A) in RPMI1640 medium containing 2% FBS for 30 min at 37 °C. Then the PD-1 NFAT-luc/Jurkat T cells (5 × 10^5^ cells/40 μL/well) in RPMI1640 medium containing 2% FBS were added to each well and co-cultured for 6 h at 37 °C. When co-cultured, inhibitor antibodies block the interaction of PD-1 and PD-L1, resulting in increasing NFAT-mediated luciferase activity in PD-1 NFAT-luc/Jurkat T cells. After the co-culturing, 80 μL Bio-Glo reagent was added and incubated for 7 min at room temperature (RT). The luminescence was measured using a microtiter plate reader (VICTOR X4, Perkin Elmer).

### 2.10. In Vivo MC38 Murine Tumor Models and Treatments

C57BL/6 mice were purchased from Orient Bio (Gyeonggi, Korea) and maintained under specific pathogen-free (SPF) conditions. All animal care and experiments were performed in accordance with protocols approved by the Institutional Animal Care and Use Committee (IACUC) of Kangwon National University (approval no. KW-171204-1). First, 5 × 10^5^ MC38 tumor cells were subcutaneously (s.c.) injected into the flanks of C57BL/6 mice. When the average estimated tumor volume reached around 80 mm^3^ ± 20 (tumor volume = width^2^ × length × 0.5), the mice were randomly divided into five groups (*n* = 5 mice per group), and three doses of 200 µg (10 mg/kg) of anti-PD-L1 antibodies (MPDL3280 and #70) or PBS control were injected intraperitoneally (i.p.) every three days. Tumors were measured 2–3 times weekly. Mice were sacrificed when the tumor volume was ≥2000 mm^3^ and tumors were surgically isolated from tumor-bearing mice.

### 2.11. Statistical Analysis

Data collected from ELISA, flow cytometry, BLI, and the cell-based assay were plotted and analyzed using IBM SPSS statistics version 24.0 (IBM Corp., Armonk, NY, USA). For a correlation analysis between binding activity and inhibitory activity, we used the Pearson correlation analysis. The correlation between binding activity and inhibitory activity was confirmed by calculating the Pearson correlation coefficient. Data from in vivo experiments were analyzed with Prism (version 5.0, GraphPad software, La Jolla, CA, USA). Two-tailed Student’s *t*-tests were used to compare each of the antibody treatment groups with a single control. In analyses of in vivo data, the error bars represent standard deviation (SD).

## 3. Results

### 3.1. Isolation of PD-L1 Targeting Antibodies from Phage-Display Library

To isolate antibody clones that bind to the PD-L1 extracellular domain, a human naïve scFv antibody library was used for the panning against the PD-L1 antigen. The panning was performed for a total of four rounds with enrichment of scFv clones against PD-L1 proteins. Successful enrichment of the antibody binders is evidenced by the apparent increase of phage recovery (output phages/input phages) at fourth panning (Appendix A) and the subsequent ELISA results reconfirmed it. After four rounds of panning, several phage-infected bacterial colonies were cultured to harvest scFv proteins using IPTG induction and applied to primary ELISA screening.

To screen out PD-L1 binding antibodies, random 1800 scFv clones from the third and fourth panning output pools were cultured, and their periplasmic extracts were harvested and applied to primary binding assays. Because of the complex contents in periplasmic fraction and the difficulties of expression normalization of individual clones, primary antibody assays were performed in a qualitative way. The initial ELISA showed signals from about 400 PD-L1-specific clones in comparison with control human Fc and BSA proteins, and the following sequence analysis confirmed 72 independent clones with different CDR sequences at heavy and light chains (Figure 1A).

To test each clone’s binding ability to the native PD-L1 structure, a FACS analysis using HEK293T cells transfected with the PD-L1 gene was performed using scFv periplasmic fractions of the selected clones (Figure 1B). Although there were small differences in binding activity, most of the scFv clones showed binding to the native PD-L1 conformation on the cell surface.

### 3.2. The Development of a BLI-Based PD-1/PD-L1 Binding Inhibition Assay

In order to develop the earlier functional assay for screened antibodies even in a periplasmic scFv format, we developed a BLI (biolayer interferometry)-based PD-1/PD-L1 binding inhibition assay using ForteBio’s Octet system. The PD-1 protein can be immobilized on the biosensor surface and allowed to interact with the analyte PD-L1 protein with or without scFv antibodies. When PD-1-coated sensors interact with the analyte PD-L1 protein in solution, a specific sensorgram can be generated; however, in theory, when the inhibitory antibody is coincubated with PD-L1, it will show changes in the sensorgrams and the degree of the changes can represent the functionality of antibody inhibition. In the assay, the PD-1 antigen was covalently immobilized on the AR2G biosensor chip via an EDC/NHS-mediated reaction as per the manufacturer’s recommendations (Figure 2A).

To set up the conditions for maximizing signals, the factors for the BLI signals for the coated protein and analyte protein, PD-1 and PD-L1, such as the pH for the coupling reaction, incubation time, PD-1 antigen concentration for the coupling reaction, and PD-L1 ligand concentration in reaction solution, were optimized (Figure 2B, Appendix A).

Overall, the pH alterations did not affect the sensorgrams showing the PD-1‒PD-L1 interaction (Appendix A), and about 1800 s of PD-1 coupling incubation on the sensor chip was enough to get saturated bindings from all the samples (Appendix A). With regard to the optimum protein concentrations, diverse combinations of PD-1 and PD-L1 contents were tested for the maximum sensorgrams. Interestingly, the PD-1 concentrations coated on the sensor chip did not affect the sensorgram patterns in the 5‒20 μg/mL range; however, there were higher peaks at higher concentrations of PD-L1 (Figure 2B).

### 3.3. The Primary Screening of Functional scFv Antibodies Using a BLI-Based PD-1/PD-L1 Binding Inhibition Assay

In order to verify the inhibitory functions of antibody clones, not just specific bindings on PD-L1 through ELISA and FACS assays, a BLI assay using periplasmic scFv antibodies was performed. Based on the parameters from the condition optimization results, PD-1-coated sensors and PD-L1 analytes were analyzed for binding on the Octet system. In the inhibition analysis, PD-L1 analytes mixed with scFv containing periplasmic extracts (1:1 volume mix) were applied and their sensorgram changes were monitored for 72 individual clones. As a negative control, an irrelevant scFv clone, specific for mucin1, isolated from a parallel phage-display panning experiment, was used to confirm the feasibility of the BLI assay. The results confirmed that a significant number of antibodies showed decreased sensorgram patterns, which indicated the inhibitory function of candidate antibodies (Figure 3), while the negative control scFv did not show the decreased sensorgram.

The inhibition efficacy (%) was calculated via the following equation:Inhibition efficacy (%) = (Rcontrol − RscFvs)/Rcontrol × 100(1)
where R_control_ and R_scFvs_ are the peak values of sensorgrams of the association phase for the PD-L1-only sample and PD-L1 plus scFv coincubation sample, respectively. After the BLI-based functional analysis, we were able to categorize the functional scFv clones into three groups: (1) High inhibition (≥30% inhibition), nine clones; (2) moderate inhibition (≥15% inhibition); 23 clones; and (3) low inhibition (<15% inhibition), 40 clones. The high and moderate inhibition groups comprised 32 scFv clones (44.4% of total 72 clones) and showed meaningful inhibition between PD-1 and PD-L1, whereas 41 scFv clones (55.6%) showed relatively marginal inhibitions (Table 1).

These results suggest that the BLI assay is highly informative for the inhibitory function assessment at the earlier antibody characterization step, even with unpurified periplasmic scFv antibodies.

### 3.4. The Comparison of the Functionality Assessment Between BLI-Based Functional Assay and In Vitro Cell-Based Assay

To confirm the inhibitory functions of the screened antibodies, a total of 72 scFvs from the initial screening were converted to IgG format, in which the VH and VL domains of individual scFv were transferred to mammalian full IgG expression vectors. Individual variable domains were PCR-amplified and then subcloned into two vector systems, pCEP4-VH and pCEP4-VL, which are for the expression of heavy and light chains of the IgG1 antibody. Paired heavy and light chain expression vectors were transfected into 293F cells at a 2:1 (light vs. heavy) DNA ratio using the FectoPRO transfection reagent (Polyplus-transfection) as per the manufacturer’s instructions. After 12 days of culture, supernatants of cell culture were harvested, and fully assembled IgG antibodies were purified by Protein A column chromatography.

To evaluate the relative functionalities of IgG format antibodies, Merck’s (Kenilworth, NJ, USA) anti-PD-1 antibody (MK3475) and Genentech’s (South San Francisco, CA, USA) anti-PD-L1 antibody (MPDL3280A) were generated as control antibodies in the assays. Expressions of each IgG antibody were briefly tested by dot blot analysis using their culture supernatants (Appendix A), in which some of the IgG convertants showed a very low expression level (15 clones). After 12 days of culture, each antibody clone was harvested and purified. After purification, 40 IgG clones showed antibody concentrations as ≥100 μg/mL, so we applied them to the PD-1 and PD-L1 blockade assay (Promega). This is a cell-based in vitro assay that can confirm the inhibition of the PD-1‒PD-L1 interaction through increased luminescence, for which PD-1 expressing Jurkat T cells with the luciferase reporter gene driven by the NFAT response element is co-incubated with PD-L1 and T cell activator-expressing CHO-K1 cells.

When purified PD-L1 antibodies were co-incubated with these two cells, they restored TCR activation signaling by blocking the PD-1‒PD-L1 interaction, and showed NFAT-driven luminescence. For the measurement of the inhibitory functions of each clone, the relative luminescence (%) was based on the luminescence value (100%) of the control, the MK3475 antibody (anti-PD-1 antibody from Merck). The assays for the 40 available clones showed that the clones had diverse inhibition levels and could be grouped into: 1) High inhibition (>50%), eight clones; 2) moderate inhibition (≥30%), 15 clones; 3) low inhibition (<15%), 17 clones. Interestingly, two clones showed a fairly high level of inhibition, #50 (91%) and #70 (89%), which were comparable to the levels of the control anti-PD1 antibody, MK3475 (100%), and the anti-PD-L1 antibody, MPDL3280A (77%) (Figure 4).

To determine whether the BLI-based functional assay has a correlation with the in vitro PD-1 and PD-L1 blockade results, we compared the results of each assay (ELISA, FACS, and BLI) with the in vitro cell-based assay. However, certain antibodies were not available for the in vitro cell-based assay due to problems in their expression in a mammalian system, so the correlation analyses were tried for 40 scFv clones that were tested in the in vitro cell-based assay, but not all 72 clones.

The applicability of the BLI-based assay with regard to the antibody’s functionality screening was compared with the results of ELISA and FACS in Figure 4. When the top 30% (red) and bottom 30% (blue) from each assays’ results were selected and compared with the clone ranking for the cell-based assay, the distributions of the top 30% and bottom 30% showed indistinguishable distribution over the clone ranking from the cell-based assay, which means that the direct binding signals from ELISA and FACS do not represent the inhibition functionalities of the antibody clones.

However, the BLI assay results showed a better distinction between the top and bottom 30%: The top 30% of clones showed higher inhibitory function in the cell-based assay (Figure 4), which showed its usefulness for the functional assessment of isolated antibodies and as a primary screening tool in the phage-display process before the laborious IgG conversion and purification steps. However, because there is no clear-cut point that differentiates functional and nonfunctional antibodies in BLI assay ranking, it could be difficult to note the specific clone above which clones in the ranking could proceed to the next development stage, such as IgG conversion. When all 72 clones are considered (Appendix A), the top 30% covered 13 clones out of 20 highly inhibitory clones from the in vitro cell-based assay (Figure 4).

### 3.5. Correlation Analysis Between Binding and Inhibitory Activities of Quantified IgG Antibodies to PD-L1

To investigate whether the analysis results from the scFv format can be translated to the characterization of quantified IgG antibodies, we performed ELISA, FACS, and BLI assays for IgG format antibodies and performed a correlation analysis with the in vitro cell-based assay results.

For IgG-format antibodies, a higher correlation was observed with the FACS assay (Pearson correlation (*r*) = 0.931, *p*-value (*p*) = 3.13 × 10^−18^) between cell surface binding and inhibitory activity in the cell-based assay, which was better than the ELISA or BLI-based assays (Figure 5). Considering that FACS and ELISA experiments are highly affected by the antibodies’ affinity, stability, and specificity toward their targets, and not always associated with the functional epitope, the clones with a high ranking in our FACS experiment must have had strong bindings to the functional epitopes. The BLI-based assay also showed a high correlation (*r* = 0.772, *p* = 1.14 × 10^−8^) with the cell-based assay results, which validated the benefit of the BLI-based inhibition assay in the functional assessment and as a primary screening tool for the functional antibody screening by phage display. Interestingly, almost every top-ranking clone from the FACS assay also showed good inhibitory function in the BLI-based assay. However, in the case of ELISA (*r* = 0.510, *p* = 7.66 × 10^−4^), some antibody clones showed a poor correlation between binding and inhibitory activity, resulting in a mixed distribution. The top and bottom 30% clones from the ELISA assay did not show higher or lower functionalities, which indicated that mere antibody‒antigen binding in the ELISA assay does not represent the functional features of antibodies. Interestingly, #50 and #70, which showed good binding activity in ELISA and FACS, also showed high inhibitory activity in the BLI-based PD-1/PD-L1-binding inhibition assay and the PD-1/PD-L1 blockade bioassay (Figure 5).

### 3.6. In Vivo Syngeneic Mouse Model Study for the Cancer Growth Inhibitory Function of a PD-L1-Targeting Antibody Candidate

To evaluate whether the functionality of the BLI-based approach could be translated to in vivo functionality, the in vivo anticancer efficacy of the selected antibody was tested through a syngeneic mouse model. Since it is well known that immune-checkpoint inhibitors can be tested only in a syngeneic mouse model where T cell immunity is intact [30,31], MC38, a murine colon adenocarcinoma, and C57BL/6 mouse models were adopted for the functional evaluation of the candidate antibody. Of the top 10 clones, #70 was selected because it showed a strong inhibitory function in the BLI and in vitro cell-based assays and strong binding signals from the FACS and ELISA assays. The fact that it showed mouse PD-L1-positive signals in the ELISA assay also made it an in vivo test candidate. We confirmed clone #70’s cross-species activities with a FACS analysis that showed strong binding to both human and mouse PD-L1 on the cell surfaces (Figure 6A). After three 10 mg/kg treatments, #70 showed a significant tumor growth inhibition effect in the MC38 tumor model, together with the control antibody MPDL3280A (Figure 6B).

Collectively, these results suggest that the clone selected through functional assays proved its in vivo functionality through the inhibition of the PD-L1 interaction in tumor cells, and that BLI-based assays have contributed to the successful screening of highly functional candidate antibodies.

### 3.7. The Application of BLI-Based Assays to Diagnostic Antibody System Development

To extend the utilization of the BLI-based inhibition assay as a general tool for antibody screening, we looked at the results of two special IgG clones in the BLI-based inhibition assay. Two clones, #41 and #71, showed high signals in ELISA (OD_450nm_; #41: 1.537, #71: 1.850, Figure 7A), and had abnormally increased sensorgrams in BLI-based PD-1/PD-L1-binding inhibition assays (Figure 7B), which suggested that these antibodies may have distal, compatible binding epitopes from the PD-1 binding site on PD-L1. The increased sensorgrams represent the binding of increased mass of the PD-L1 plus antibody complex to PD-1 on the sensor chip. To confirm this hypothesis, we tested the binding compatibilities of each clone with an MPDL3280A control antibody that has the same binding epitope with PD-1 through the competitive ELISA and BLI assays, for which direct competition between PD-1 and each clone would be difficult because the binding between PD-1 and PD-L1 (0.77 µM) is weaker than the common antigen‒antibody interaction [32].

The results of both assays showed that most clones did not have binding compatibility with MPDL3280A, while clones #41 and #71 clearly showed compatible binding signals with MPDL3280A in the competitive ELISA (Appendix A) and BLI-based binding compatibility assays (Figure 7C), which means that the clones with increased sensorgrams in the BLI-based PD-1/PD-L1 binding inhibition assay can be categorized as ones with compatible binding with PD-1. So, through a BLI-based inhibition assay, we were able to categorize the anti-PD-L1 antibodies into two groups with regard to their binding compatibilities with PD-1, which can be applied to the screening of at least two groups of antibodies. This is especially important for the development of the diagnostic antibody pairs needed for sandwich ELISA-type antigen detection.

## 4. Discussion

Therapeutic antibodies have become some of the most promising drugs for diverse diseases in the clinic. So far, 93 monoclonal antibodies have been FDA approved for therapy, and a number of candidates are in the advanced phases of clinical trials [33,34].

Phage display has been a fundamental, robust, and well-established technology for the development of therapeutic antibodies. In the process of phage display, the initial screening of antibodies has usually been performed by an ELISA assay using scFv, Fab, or the phage particle itself; a sequencing analysis follows to identify independent clones. Usually these independent clones are transformed into IgG-format antibodies for the functional characterization. When there are many independent clones, the transformation to the IgG format itself can be highly difficult work to most of the academic/industry researchers. Although the first ELISA assay can only show each antibody’s basic target binding, the following assays with IgG antibodies are related to the functional assessment of the clones, such as the verification of the target’s function change with other molecules, such as the receptor‒ligand interaction by antibody treatment. Therefore, if the assessment of the inhibitory function of the scFv antibody is possible in the earlier phase, prior to full IgG conversion, it would be highly beneficial to the whole process of developing therapeutic antibody candidates.

BLI-based assays have been very useful tools and the instrument has been a flexible high-throughput system for the antibody development process. It has been well used as a screening tool for high-level antibody-producing cell-line development [35] and initial antibody selection [36]. As an initial antibody screening tool for the hybridoma antibody selection, Lad et al. (2015) reported the strong benefit of the high-throughput kinetic screening of hybridoma antibodies, focusing on the dissociation rate (off-rate) using BLI technology [36], which effectively screened out high-affinity antibodies that could be ignored by typical ELISA-based hybridoma antibody screening. However, this screening method still cannot verify the functionality of isolated antibodies from the screening unless there are additional functional assays.

In our study, we developed new BLI-based PD-1/PD-L1 inhibition assays that can be applicable to periplasmic scFv antibodies, in which the direct interaction between target (PD-L1) and receptor (PD-1) can be measured first; then any sensorgram changes from the addition of the antibody candidate to the target and receptor interaction are compared. Our study focused more on the assessment of the functionality of antibody clones from the phage-display antibody screening, which indicated the inhibition of the target protein’s (in our case, PD-L1) interaction with its partner (PD-1). With the help of an innovative instrument, the Octet RED96 system from ForteBio, we were able to set up the experimental conditions using a 96-well-type inhibition assay for the PD-1‒PD-L1 interaction. When we applied the assay, even to the unnormalized periplasmic scFv antibodies, PD-L1 incubated with antibodies clearly showed the reduced sensorgrams that represented the inhibitory functions of antibody clones, while the negative control antibody did not show such a sensorgram change.

In comparison with the results from the cell-based inhibition assay with IgG antibodies, the scFv clones with high inhibition (the top 30% group) were mostly included in the set of highly functional antibodies in the cell-based assay, in contrast to the high scFv binders from the ELISA or FACS assay, which means that the BLI-based functional assay can be very informative in candidate selection and increase the probability of functional antibody development at the initial screening stage. When the correlation between the cell-based assay and ELISA/FACS/BLI-based assay with scFv antibodies was analyzed, there was no statistically meaningful correlation, which could be because of the unnormalized scFvs, which is usually the main problem with primary antibody screening. It is difficult to make scFv expression even in the screening when there are so many clones in ELISA and other assays. However, IgG-format antibodies showed a relatively high level of correlation between the cell-based assay and the ELISA/FACS/BLI assay. In particular, the FACS and BLI-based assays showed strong correlations with the cell-based assay (*r* = 0.931, *p* = 3.13 × 10^−18^, and *r* = 0.772, *p* = 1.14 × 10^−8^), which confirmed the utility of BLI-based inhibition assays in functional antibody development.

As for the higher correlation of the FACS assay with the in vitro cell-based assay in our IgG case, in theory, the high FACS shift represents the high-affinity antibody binding to the overexpressed antigens or well-exposed epitopes in a native conformation, but such a high FACS shift does not always prove the functionality of the epitopes. So, it is a very interesting result from our experiment, but it cannot be generalized so as to claim that the FACS assay has in most cases a strong correlation with the in vitro cell-based functional assay. However, for the BLI assay, the sensorgram shows the relative bindings between target protein (PD-L1) and counterpart (PD-1) with or without the inhibitory antibodies, which in turn represent the functionality of the antibodies on the target’s interaction. So, even though its correlation is slightly weaker than in the FACS assay in the current experiment, the fact that the BLI assay has a correlation with the in vitro functional assay itself is an important validation of the feasibility of using the BLI-based assay as a functional assay tool in therapeutic antibody development. As to the higher correlation of the FACS assay with an in vitro assay in our results, we reasoned that our phage-display panning against PD-L1 had generated, in some way, many antibodies targeting the epitopes that overlapped with the PD-1 binding site and so was very sensitive on their conformation (the ELISA results did not show such correlations).

When we looked at the covariation between the ELISA, FACS, and BLI-based inhibition assays, there were no statistically meaningful covariations, which means they showed different features of scFv antibodies at the first screening step after phage display and the BLI-based assay was important to evaluate the functionality of the antibody clones. Interestingly, a lot of low-inhibition scFv clones showed a low level of expression in mammalian systems, so most of them could not be tested for a cell-based assay. This could be because the low inhibition activities of scFv come from nonfunctional properties or the low expression of individual clones; however, we cannot conclude which factor contributed most.

For testing in vivo antibody activity, especially for the PD-L1-targeting antibody, it was difficult to compare the BLI assay to in vivo functionality because only human‒mouse cross-species reactive antibodies are eligible for an in vivo assay. Human‒mouse cross-reactive clone #70 was chosen as an in vivo test clone. Moderate inhibition activity in a BLI assay using scFv was selected for the in vivo animal model study and showed strong anticancer inhibition activity, together with Genentech’s MPDL3280A control antibody. To become a final therapeutic antibody candidate, further analysis of various parameters such as the stability, expression, and solubility of the antibody will be needed.

Clones #41 and #71 showed very interesting results: BLI-based sensorgrams showed increased peaks compared to the control sensorgram. Later, through the competition ELISA and BLI assays, they showed compatible binding to PD-L1, together with MPDL3280A, which means their epitopes are relatively far from the control antibody’s binding site. So, the clones showing these increased sensorgrams can be distinguished from the other clones and this categorization of two antibody groups with regard to their binding compatibility with the native ligand (PD-1) can be useful to the screening of at least two groups of antibodies for a sandwich-type antigen capture system or a multiple epitope targeting system.

The BLI-based functional assay has several advantages in phage-display screening: (1) It can be easily applied to a high-throughput screening system, so an automated functional assay at primary antibody screening in phage display can generate a highly efficient antibody screening process; (2) the qualitative assessment of functional antibody development from phage panning can be analyzed even at the initial scFv screening stage, and a BLI-based assay with an IgG antibody will provide highly reliable data for the development of therapeutic antibodies; and (3) it will be a highly efficient method to develop diagnostic antibody pairs for companion diagnostics.

Overall, we designed a new BLI-based functional assay for initial scFv screening and tried it out to screen functional antibodies’ PD-L1 targeting. A BLI-based assay in earlier stages showed a higher functional correlation with a later in vitro cell-based functional assay. That result correlated well with the results of an in vivo syngeneic mouse model study. Our study showed that the BLI-based assay is highly beneficial for functional antibody screening at earlier steps and for drug candidate development.

## Figures and Tables

**Figure 1 viruses-12-00684-f001:**
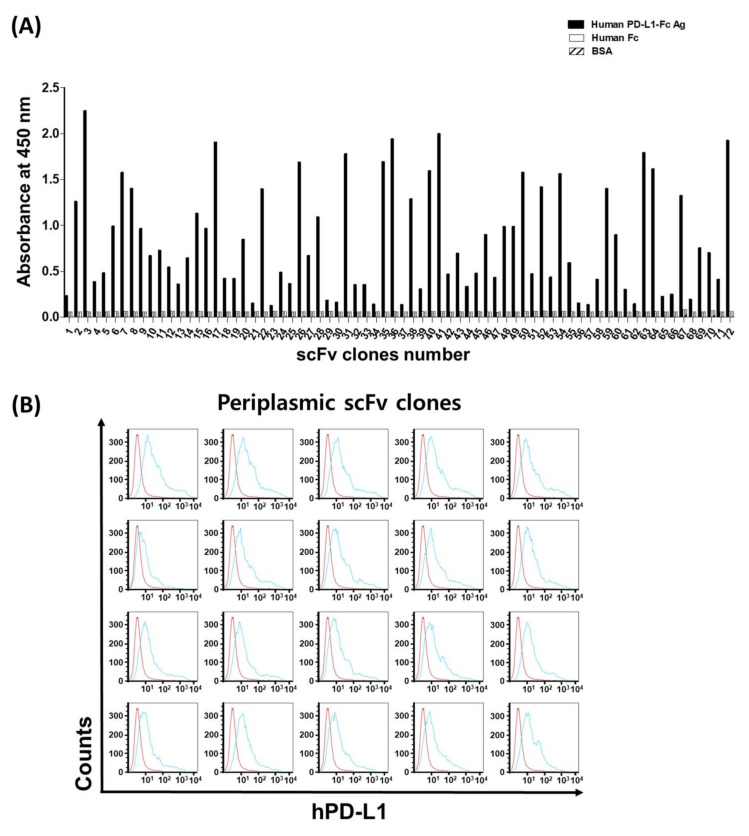
Isolation of scFv clones specific to human PD-L1s. (**A**) The individual specific binders are screened by ELISA and 72 different scFv clones were further characterized by sequencing. Reactivity of the selected scFv clones to human PD-L1 was assayed by measuring absorbance at 450 nm. The individual scFv clones were reactive to hPD-L1-Fc (■), but not hFc (□) or BSA (▨); (**B**) Binding activity of scFv (top 20 clones) on hPD-L1 293T cells measured using flow cytometry. 293T cells were transfected with human PD-L1s. Binding of periplasmic scFvs was detected by adding first anti-c-Myc murine monoclonal antibody followed by Alexa Flour 488-conjugated antimurine IgG (H+L).

**Figure 2 viruses-12-00684-f002:**
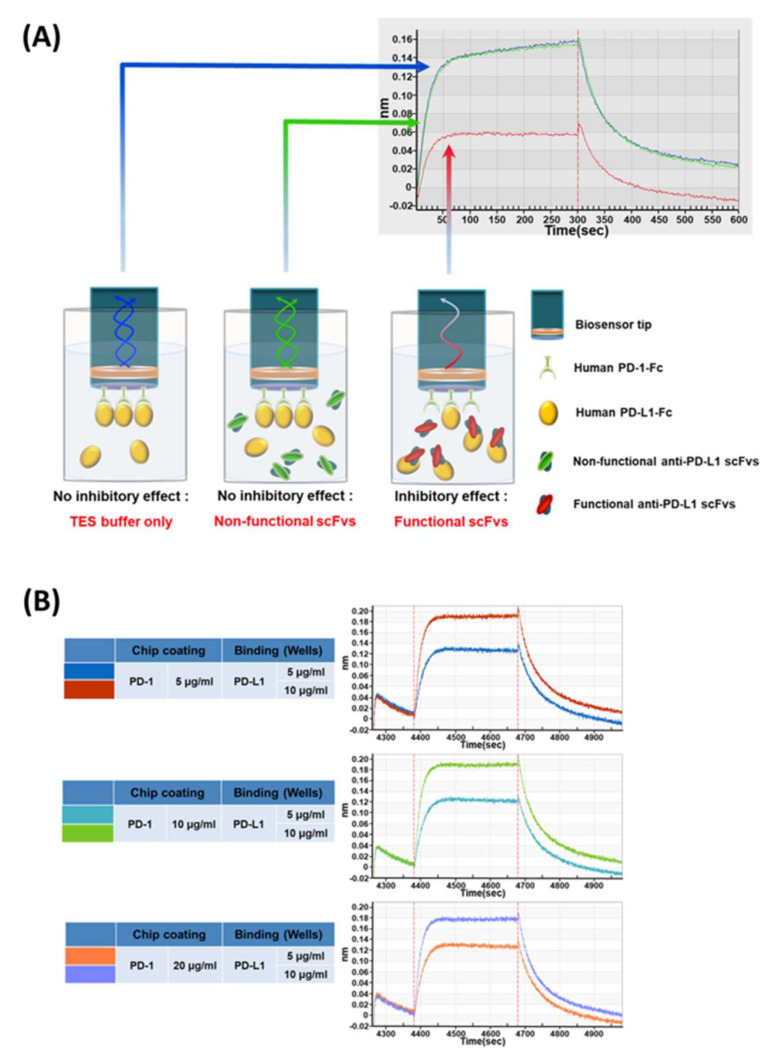
Illustration of biolayer interferometry (BLI)-based assay and optimization for functional antibody screening. (**A**) The AR2G biosensors were incubated with human PD-1-Fc. The PD-1-Fc immobilized AR2G biosensors were blocked using 3% BSA, followed by incubation with human PD-L1-Fc in the absence or presence of periplasmic scFvs. A binding inhibition signal was detected by BLI, and the inhibition signal intensity of periplasmic scFv antibodies was calculated by normalizing the PD-1/PD-L1 binding signal to 100% in the absence of periplasmic scFv antibodies; (**B**) Optimization of PD-1/PD-L1 protein binding conditions in BLI. The binding capacity and concentration. Assessment of BLI response signal (nm) for human PD-1/PD-L1 protein binding on AR2G chip. Human PD-1 protein was immobilized onto AR2G biosensors via amine coupling.

**Figure 3 viruses-12-00684-f003:**
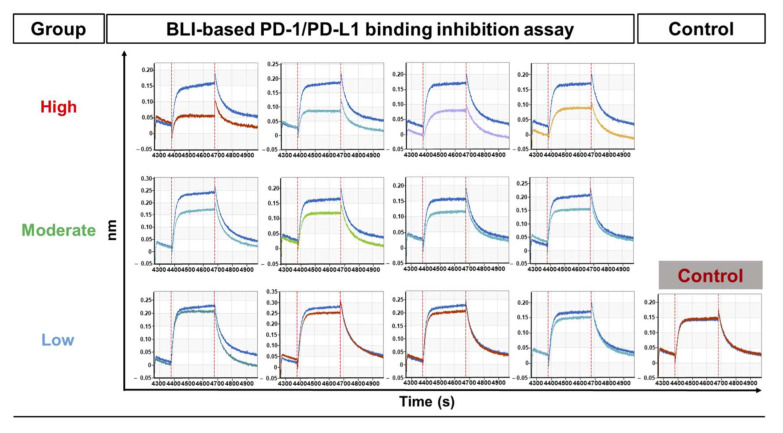
Screening of functional scFv antibodies using an optimized BLI-based PD-1/PD-L1-binding inhibition assay. Inhibitory activity analysis of 72 individual periplasmic scFv antibodies. Analysis of inhibitory activity against PD-1‒PD-L1 interaction was performed using BLI-based assays under optimized conditions. The relative inhibitory efficiency of scFvs was calculated by normalizing the PD-1/PD-L1-binding response in the presence of the TES buffer to 100%. Classification and response plots (sensorgrams; for the top four clones of each group) of the three groups by difference in inhibitory efficacy (high, ≥30%; moderate, ≥15%; and low, <15%) are presented. For the negative control, the anti-mucin1 antibody isolated from the parallel antibody screening campaign was used. It did not show any interferences with the PD-1‒PD-L1 interaction.

**Figure 4 viruses-12-00684-f004:**
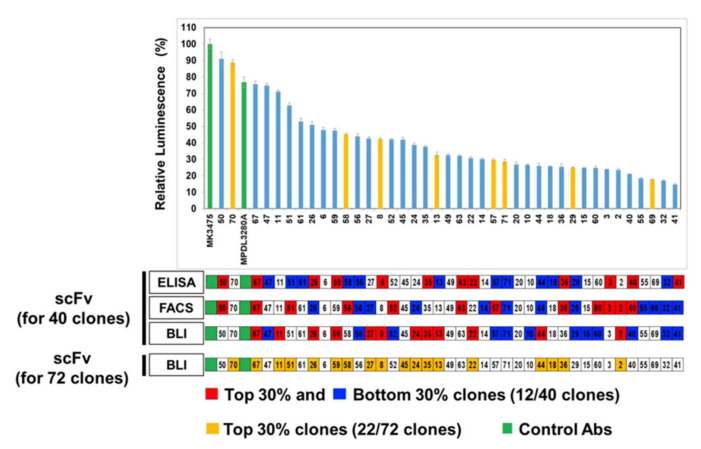
In vitro cell-based PD-1/PD-L1 inhibition assay in comparison with primary scFv screenings. Evaluation of in vitro assays to assess the inhibition of PD-1‒PD-L1 interactions by full IgG antibodies. The activity of 40 full IgG converted antibodies in blocking the effect of the PD1/PD-L1 checkpoint on TCR-mediated T cell activation is assessed as the level of luciferase activity. MK3475 (anti-PD-1) and MPDL3280A (anti-PD-L1) antibodies were used as positive controls (green bar). The light blue bars have binding activity of antibody clones against human PD-L1 antigen, whereas yellow bars have cross-reactivity to both human and mouse PD-L1 antigens. Results are classified using color shading codes, with the top 30% in red and the bottom 30% in blue in each assay (scFvs 30%; 12/40 clones and 22/72 clones).

**Figure 5 viruses-12-00684-f005:**
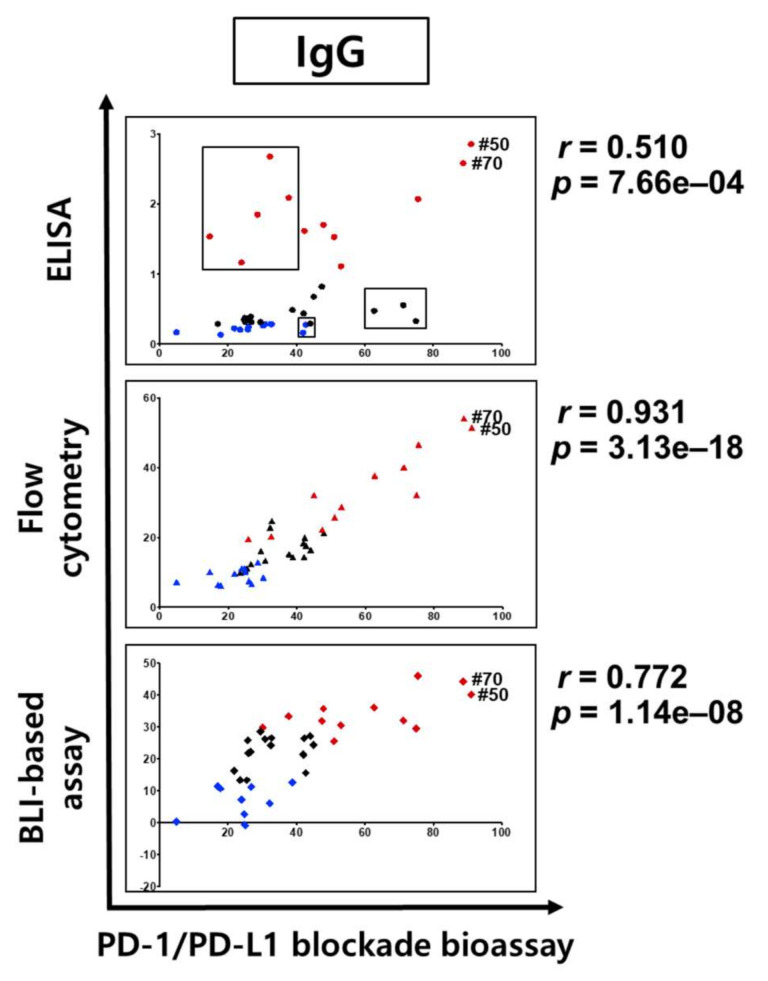
Correlation analysis of binding and inhibitory activities of IgG format antibodies. Comparative analysis of binding specificities and in vitro activity according to antibody formats. The vertical axis is the binding and binding inhibition activity in ELISA (circle, 450 nm), flow cytometry (triangle, MFI; mean fluorescence intensity), and BLI-based PD-1/PD-L1 binding inhibition assay (diamond, inhibition %), and the horizontal axis is the PD-1‒PD-L1 interaction inhibitory activity (relative luminescence %) of the antibodies. For the BLI-based PD-1‒PD-L1 binding inhibition assay using IgG, clones #41 (‒144%) and #71 (‒37.63%) are not indicated. Poorly correlated clones are represented by three black boxes. The Pearson correlation coefficient (*r*) and *p*-value (*p*) are given in the top panel (*r* = 0.510, *p* = 7.66 × 10^−4^), middle panel (*r* = 0.931, *p* = 3.13 × 10^−18^), and bottom panel (*r* = 0.772, *p* = 1.14 × 10^−8^). Results are classified using color shading codes, with the top 30% in red and the bottom 30% in blue (IgGs; 30%; 12/40 clone) in each of the three assays (ELISA, flow cytometry, and BLI-based PD-1/PD-L1-binding inhibition assay).

**Figure 6 viruses-12-00684-f006:**
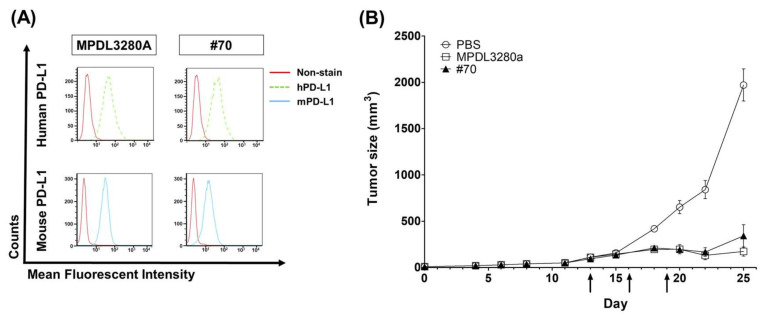
In vivo function of anti-PD-L1 antibody in MC38 syngeneic mouse model. (**A**) Flow cytometry analysis demonstrates cross-reactive binding of IgG-converted anti-human PD-L1 antibodies to human PD-L1 in PC3 cells, and mouse PD-L1 in mouse IFN-γ-treated MC38 tumor cells (100 ng/mL for 24 h); (**B**) Tumor growth curves of C57BL/6 mice subcutaneously injected with MC38 tumor cells. Mice (C57BL/6 bearing MC38 tumors, *n* = 5 per group, mean ± SD) were treated with PBS or mouse PD-L1 cross-reactive antibodies (MPDL3280A and #70). Treatment was given by intraperitoneal injection (10 mg/kg) on the days marked with black arrows. *t*-test; ***; *p* ≤ 0.001 by unpaired *t*-test on day 25.

**Figure 7 viruses-12-00684-f007:**
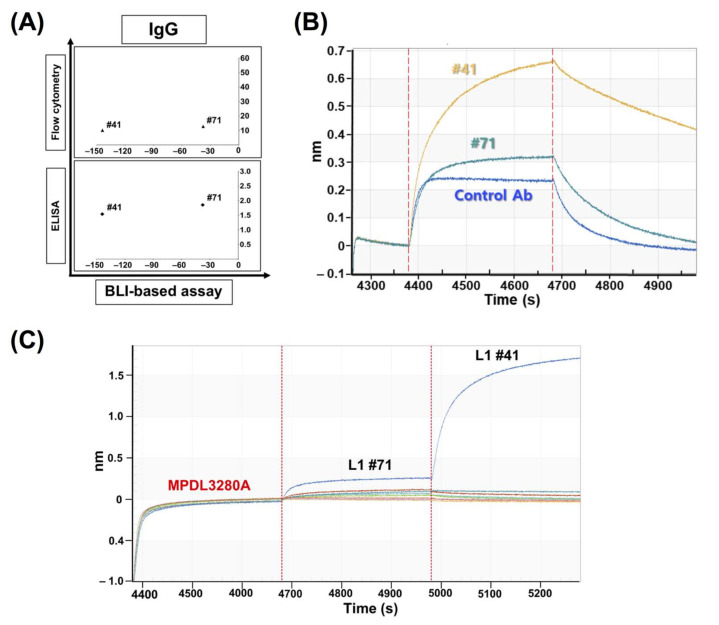
The application of the BLI-based assay for epitope characterization of antibodies. (**A**) Classification and assessment of abnormal binding sensorgrams in BLI-based PD-1/PD-L1-binding inhibition assays. In the assay, PD-1 is coated on the sensor and a mixture of PD-L1 and each IgG antibody clone was incubated with the sensor; (**B**) Distribution of #41 (‒144%) and #71 (‒37.63%) clones in BLI-based assays using IgG; (**C**) Binding compatibility test using the BLI assay. The PD-L1-coated sensors were pre-saturated with MPDL3280A antibody binding and two antibodies were introduced to the sensors consecutively to check if any clone has compatible binding with MPDL3280A (tested clones: #3, 6, 10, 24, 26, 29, 35, 40, 41, 50, 51, 52, 56, 63, 70, 71).

**Table 1 viruses-12-00684-t001:** Classification of functional scFv antibodies.

Inhibition (%)	% in Total	*n*/72 Clones
≥30%	12.5%	9
≥15%	31.9%	23
<15%	55.6%	40

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
