# Peer review of "BLI-Based Functional Assay in Phage Display Benefits the Development of a PD-L1-Targeting Therapeutic Antibody"

_viruses, 2020, doi:10.3390/v12060684_

Round 1

Reviewer 1 Report

Choi and co-workers describe the development of PD-L1 targeting antibody by biolayer interferometry (BLI) based assay technology. However, as they highlight in their manuscript, typical phage display is challenging to identify therapeutic antibody candidates. Thus, to move this field forward they needed to develop an efficient assay method for getting therapeutic candidates. They utilize a BLI based functional assay to screen functional PD-L1 targeting antibodies. BLI-based functional assay enabled to carry out high throughput assay that benefits functional antibody therapeutics when compared to previous screening methods, ELISA assay and FACS analysis. Furthermore, the authors showed that selected antibody significantly inhibited tumor growth. Finally, they applied BLI methods to characterize epitope. Generally speaking, this study is well designed and solid with the appropriate controls and experiments. Furthermore, the writing is clear and their claims are supported by the data. The authors need to address the following comments prior to the acceptance for the publication.

Minor comments:

  1. Quality of figures (Fig 1, 2, 3) is very poor such as too small letters in the figures. These issues prevented full assessment of the work and would need to be addressed prior to publication.

  2. BLI-based assay was applied to characterize epitope of antibodies, which is poorly described here. It is strongly recommended that sentences 435-446 are supposed to be well organized/re-written to deliver the results in Figure 7.

Author Response

Response to Comments and Suggestions from Reviewer 1

-------------------------------------------------------------------------------

[ Comments and Suggestions for Authors_REVIEWER 1 ]

Choi and co-workers describe the development of PD-L1 targeting antibody by biolayer interferometry (BLI) based assay technology. However, as they highlight in their manuscript, typical phage display is challenging to identify therapeutic antibody candidates. Thus, to move this field forward they needed to develop an efficient assay method for getting therapeutic candidates. They utilize a BLI based functional assay to screen functional PD-L1 targeting antibodies. BLI-based functional assay enabled to carry out high throughput assay that benefits functional antibody therapeutics when compared to previous screening methods, ELISA assay and FACS analysis. Furthermore, the authors showed that selected antibody significantly inhibited tumor growth. Finally, they applied BLI methods to characterize epitope. Generally speaking, this study is well designed and solid with the appropriate controls and experiments. Furthermore, the writing is clear and their claims are supported by the data. The authors need to address the following comments prior to the acceptance for the publication.

  1. Quality of figures (Fig 1, 2, 3) is very poor such as too small letters in the figures. These issues prevented full assessment of the work and would need to be addressed prior to publication.

Response:

We thank the reviewer for pointing out this problem. As to the reviewers' comments on the quality of the figures, especially for Figure 1, 2, 3, 4 and 6, we have revised the figures as requested.

  1. BLI-based assay was applied to characterize epitope of antibodies, which is poorly described here. It is strongly recommended that sentences 435-446 are supposed to be well organized/re-written to deliver the results in Figure 7.

Response:

We found that there were several ambiguous jumps of experiment rationales and reasonings in the description of that result paragraph for Figure 7. Also, the concepts of ‘competitiveness’ and ‘compatibility’ for antibodies in the paragraph were a bit poorly described in the original manuscript. So, we rewrote the whole result section for Figure 7 and described the results with more details and additional supplementary figure in accordance with the reviewer’s suggestion.

Reviewer 2 Report

This manuscript focuses on the use of bio-layer interferometry (BLI) to screen antibodies for functiona. The antibodies are initially identified scFv phage display library panning and subsequently cloned in expression systems, whereby the crude antibody extracts (crude scFv periplasmic extracts) are screened for function using BLI and compared to traditional ELISA and flow cytometry methods. Finally, the authors demonstrate the functionality of selected antibody clones for efficacy towards a syngeneic tumor model and compared with clinically used, commercial antibodies. The findings of the manuscript are interesting and would be of interest to the readership using viruses to identify antibodies (i.e. phage display). However, there are some major concerns that need to be addressed before it can be considered for publication. The comments are provided below in a point-by-point fashion for clarity.

Major:

  1. Can the authors distinguish their work compared to Lad et al., Journal of Biomolecular Screening (2015), “High-Throughput Kinetic Screening of Hybridomas to Identify High-Affinity Antibodies Using Bio-Layer Interferometry”? Lad et al used BLI to identify high-binding antibodies. The only main difference on surface is that this work uses antibodies identified from phage display, and Lad et al. identified from hybridomas. This is critical to address the novelty and motivation of the authors’ work compared to others. Please address this comment and provide discussion of this manuscript’s results compared to earlier work.
  2. In methods, section 2.3, more detail must be provided about the phage display library used. It also causes confusion regarding if there is a c-myc tag (for example, later results talk about probing using anti-c-myc antibodies).
  3. Related to point 2, please write some of the methods section with more detail. Think of it as we would want the readership of the journal to be able to replicate this work and use BLI as a useful screening tool. While some parts have excellent detail (e.g. ELISA), others (e.g. phage display, panning) need more detail.
  4. In lines 231-233, the authors comment on enrichment in panning with subsequent rounds but do not show the data. Please include the data, even if it is supplementary. This strengthens the argument that panning is effective.
  5. In lines 266-269, authors comment that “pH condition for the coupling reaction, incubation time, PD-1 antigen concentration for the coupling reaction and PD-L1 ligand concentration in reaction solution were optimized (Figure 2B)”. Figure 2B does not pH and incubation time optimization. Please show all relevant data as stated in sentence from lines 266-269.
  6. Lines 345-347—here the Merck and Genentech antibodies are presented. Why are the antibodies not used in the BLI data as a positive control (Figure 3). Please discuss and include that data if possible. These antibodies are good positive controls that would further help validate the BLI assay. Or why were they not included?
  7. Lines 363-365—can authors make this claim since some clones could not be incorporate into analysis due to poor expression? Please address.
  8. Flow cytometry has better correlation than BLI to identify antibodies (Figure 5). The authors should address this in the discussion to explain despite the “disadvantage” of BLI, how is more advantageous than using flow cytometry? Otherwise, one would use flow cytometry and invalidate use of BLI.
  9. This may have been missed by me, but the authors should show data and/or discuss of how the crude extracts interfere with the signal in BLI. I imagine the other components of the crude extract could confound the signal. Please address.

Minor:

  1. Line 272—present the data, even if supplementary.
  2. Line 443—it says data not shown. Show the data, even if in supplementary! The addition further strengthens the authors’ assertions.
  3. Throughout the manuscript, please re-edit to improve clarity of manuscript.

Author Response

Dear Reviewer,

Thank you,

Dae Hee Kim

Round 2

Reviewer 2 Report

The authors addressed comments. However, a few comments:

The authors do this but make sure to refer to this work as screening and not selection. This is in part due to the fact that with increasing round of panning, there was a decrease in relative output (Table S2, except for round 4). Unless the authors can cite a reference, the literature suggests that after Round 2, there should be an increase, even with stringency. So without proper enrichment, it is not a selection, but a screening. Please change text accordingly.

My main suggestion is to go back and revise the grammar of the manuscript. The clarity of grammar throughout could be greatly improved. 
